# Cryo-EM structure of the human Asc-1 transporter complex

Yaning Li[1,2,3,5], Yingying Guo[1,5], Angelika Bröer[4,5], Lu Dai[1,5], Stefan Bröer [4] ✉ &
Renhong Yan [1] ✉

The Alanine-Serine-Cysteine transporter 1 (Asc-1 or SLC7A10) forms a crucial heterodimeric transporter complex with 4F2hc (SLC3A2) through a covalent disulfide bridge. This complex enables the sodium-independent transport of small neutral amino acids, including L-Alanine (L-Ala), Glycine (Gly), and D-Serine (D-Ser), within the central nervous system (CNS). D-Ser and Gly are two key endogenous glutamate co-agonists that activate N-methyl-d-aspartate (NMDA) receptors by binding to the allosteric site. Mice deficient in Asc-1 display severe symptoms such as tremors, ataxia, and seizures, leading to early postnatal death. Despite its physiological importance, the functional mechanism of the Asc-1-4F2hc complex has remained elusive. Here, we present cryo-electron microscopy (cryo-EM) structures of the human Asc-1-4F2hc complex in its apo state, D-Ser bound state, and L-Ala bound state, resolved at 3.6 Å, 3.5 Å, and 3.4 Å, respectively. Through detailed structural analysis and transport assays, we uncover a comprehensive alternating access mechanism that underlies conformational changes in the complex. In summary, our findings reveal the architecture of the Asc-1 and 4F2hc complex and provide valuable insights into substrate recognition and the functional cycle of this essential transporter complex.

Amino acid transport plays fundamental roles in mammalian cellular metabolism, protein biosynthesis, and signal transduction, which is mediated by specific transporters across the cell membrane[1–3]. Among these transporters, heterodimeric amino acid transporters (HATs) stand out as crucial players. HATs consist of two-subunit protein complexes, comprising a heavy chain and an associated light chain, belonging to the solute carrier (SLC) families SLC3 and SLC7, respectively[4–6]. HATs serve as a critical class of amino acid transporters, responsible for shuttling various amino acids, hormones, and amino acid-like drugs across cellular membranes.

The light chains of HATs, also known as glycoprotein-associated amino acid transporters, catalyze the transport of amino acids across cell membranes[6]. The associated glycoprotein 4F2hc is one of two

known heavy chain proteins of HATs that are required for the membrane trafficking and the transport activity of the light chains of HATs, the other one being rBAT[3,7,8]. 4F2hc is found to be overexpressed in many kinds of tumor cells, and involved in multiple activities related to cell growth, cell adhesion, angiogenesis and malignant transformation[9–11]. Notably, aberrant expression and mutations in HATs can lead to diseases such as cystinuria and lysinuric protein intolerance (LPI) and are also linked to certain types of cancers[12–14].

The Asc-1 (also known as asc-1 or SLC7A10) and 4F2hc complex is a typical member of HATs, which can mediate the transport of small neutral L- and D-amino acids, such as glycine (Gly), L-serine (L-Ser), L-alanine (L-Ala), L-cysteine (L-Cys) and in particular D-Ser, across the plasma membrane[15,16]. The dominant transport mode is via exchange

[1]Department of Biochemistry, Key University Laboratory of Metabolism and Health of Guangdong, School of Medicine, Southern University of Science and Technology, Shenzhen, Guangdong Province, China. [2]Institute for Biological Electron Microscopy, Southern University of Science and Technology, Shenzhen, Guangdong Province, China. [3]Beijing Advanced Innovation Center for Structural Biology, Tsinghua-Peking Joint Center for Life Sciences, School of Life Sciences, Tsinghua University, Beijing 100084, China. [4]Research School of Biology, Australian National University, Canberra, ACT, Australia. [5]These authors contributed equally: Yaning Li, Yingying Guo, Angelika Bröer, Lu Dai. ✉e-mail: Stefan.broer@anu.edu.au; yanrh@sustech.edu.cn

although some facilitated diffusion occurs and its substrate specificity is symmetric[15–18]. In the nervous system, Asc-1 has been proposed to serve two roles, namely the release of glycine from astrocytes in the brain-stem and spinal cord and the release of D-Ser from neurons or astrocytes in the hippocampus[17,19]. The *asc-1* null mice display notably reduced levels of glycine in the spinal cord and substantially diminished spontaneous glycinergic postsynaptic currents in motor neurons[17,20], suggesting that release of glycine is the major role of this transporter. Consistently, *asc-1* knock-out mice exhibit hyperexcitability causing tremors, ataxia, and seizures, and leading to early postnatal death with a median lifespan of 21 days[21]. Using mice with a β-galactosidase gene behind the *asc-1* promoter, expression was located in astrocytes mainly in the brain-stem and spinal cord. These results suggest that glial Asc-1 mediates the release of glycine after its removal from the synaptic cleft by the glial glycine transporter GlyT1. It is then returned to glycinergic neurons via GlyT2[17].

In the cortex, Gly and D-Ser serve as two endogenous glutamate co-agonists that activate N-methyl-d-aspartate (NMDA) receptors by binding to the strychnine insensitive binding site[22,23]. Rutter et al. found greatly reduced uptake of D-Ser in synaptosomes extracted from brains of *asc-1* knockout mice[19]. Physiologically, the transporter is more likely to be involved in the release of D-Ser from neurons or astrocytes. As a result, inhibitors targeting Asc-1 could decrease the tonic release of D-Ser and impair NMDAR function, providing therapeutic benefits for schizophrenia at concentrations that do not affect its role in inhibitory neurons[24–26]. Thus, Asc-1 is a potential drug target for treating cognitive affections impairment and schizophrenia diseases[27–32].

In addition to its role in the brain, Asc-1 is highly expressed in adipose tissue. Its expression is inversely correlated with body mass index, insulin, insulin resistance, glucose, and triglycerides. This indicates that Asc-1 may be a therapeutic target to treat insulin resistance and type 2 diabetes[33]. Loss of Asc-1 in subcutaneous pre-adipocytes resulted in spontaneous differentiation into beige adipocytes[34], which could be used to reduce adiposity with Asc-1 inhibitors that do not cross the blood-brain-barrier.

Despite the crucial physiological functions of Asc-1, many essential questions about the structure and function of the Asc-1-4F2hc complex remain unanswered. While several cryo-EM structures of human HATs have been determined previously, they have not provided a clear answer to the critical question of how the Asc-1-4F2hc complex recognizes and transports substrates[7,35–44].

Here, we determine the cryo-EM structures of Asc-1-4F2hc complex bound with its amino acid substrates. Combining structural analysis with transport assays, we reveal a comprehensive alternating access mechanism that underlies conformational changes in the transporter complex.

## Results

### Structural determination of the Asc-1-4F2hc complex
To investigate the transport mechanism, we purified the Asc-1-4F2hc complex after co-expression of full-length human His-tagged 4F2hc and Flag-tagged Asc-1 in human embryonic kidney (HEK) 293 F cells. The membrane fraction was solubilized using 25 mM Hepes (pH 7.5), 150 mM NaCl, 1% Lauryl maltose neopentyl glycol (LMNG) supplemented with 0.1% cholesteryl hemisuccinate Tris salt (CHS), followed by tandem affinity purification in modified solubilization buffer containing 0.01% LMNG and 0.001% CHS. During subsequent size exclusion chromatography (SEC) in 25 mM Hepes (pH 7.5), 150 mM NaCl, and 0.02% Glyco-diosgenin (GDN), the complex exhibited a single, well-defined peak and its quality was confirmed through Coomassie blue staining on sodium dodecyl sulfate–polyacrylamide gel electrophoresis (SDS–PAGE) gels, indicating a high level of homogeneity (Fig. S1a).

We then determined the cryo-EM structures of the Asc-1-4F2hc complex in three distinct states: the apo, a D-Ser bound, and a L-Ala bound, at an overall resolution of 3.6 Å, 3.5 Å, and 3.4 Å, respectively (Fig. 1a, b and Fig. S2). Detailed information related to cryo-EM sample preparation, data collection and processing, as well as model building, can be found in the "Materials and Methods" section, Figs. S1, S3, S4 and Table S1.

The HAT-specific disulfide bond is found between Cys154 of Asc-1 and Cys211 of 4F2hc (Fig. 1a, b and Fig. S2). Overall, both the apo and D-Ser-incubated structures exhibit an inward open conformation, with 12 transmembrane (TM) helices forming the characteristic LeuT-fold (Fig. 1a and Fig. S2). An additional density with a shape consistent with a D-Ser molecule was observed in the cryo-EM map of Asc-1-4F2hc incubated with D-Ser (Fig. 1a), but was absent in the apo state (Fig. S2), supporting the presence of D-Ser. Similarly, the density corresponding to L-Ala was also identified in the cryo-EM map of Asc-1-4F2hc + L-Ala (Fig. 1b). However, the structure of Asc-1-4F2hc + L-Ala adopts an occluded conformation (Fig. 1b).

In addition to the disulfide bridge, 4F2hc interacts non-covalently with Asc-1 at the extracellular side, intracellular side, and transmembrane region (Fig. 2), akin to the structure of the LAT1-4F2hc complex, another representative member of HAT[7]. Nevertheless, upon closer examination of the interaction mechanism, significant differences emerge. Apart from the conserved disulfide bond, Asn153 and Gln294 of Asc-1 could form hydrogen bonds with Arg535 and Lys533 of 4F2hc, respectively, in the apo state (inward open Fig. 2b). When L-Ala is bound, a significant conformational change occurs in Asc-1, disrupting the hydrogen bond between Gln294 and Lys533 and the intracellular interface is also remodeled (Fig. 2c). Conversely, in the LAT1-4F2hc structure, Lys533 might form a salt bridge with Glu303 of LAT1, while Arg535 forms hydrogen bonds with Thr163 and Gln304 of LAT1 in the inward facing state, indicating a stronger interaction compared to the Asc1-4F2hc complex[7].

### Substrate binding patterns of Asc-1
The precise poses of D-Ser and L-Ala could not be determined at this resolution. Based on the analysis of the interaction between substrate and Asc-1, as well as the similarity of substrate-binding pockets among the HAT family homologous proteins[38,41], we built L-Ala and D-Ser in the density of substrates. The binding patterns of D-Ser and L-Ala differ slightly in the Asc-1-4F2hc complex, due to the different conformation adopted by the transporter (inward open vs occluded) (Fig. 1c). As a result, L-Ala is positioned closer to the unwound region of TM1 than D-Ser (Fig. 1c, d). In the D-Ser bound complex, the α-carboxylate group of D-Ser potentially forms hydrogen bonds with the main chains of Ser56, Gly57, Phe243, and Ala244, while the α-amino group is stabilized by the carbonyl oxygen atoms of Asn52 and Ile53. Notably, the side chain of substrate D-Ser may form hydrogen bonds with the side chain of Tyr333 in TM8 (Fig. 1c). By comparison, substrate L-Ala is more likely to be stabilized by the side chains of Tyr333 and Ser56, as well as the main chains of Asn52, Ile53, Gly57, Phe243, and Ala244.

The transport path of Asc-1 can be readily identified by comparing the D-Ser (inward open) and L-Ala bound (occluded) conformations (Fig. 3a). Notably, Glu257 in TM6b and Arg339 in TM8 might form polar interactions in the occluded conformation, which are disrupted in the inward facing state. From these two residues onwards the transport path is lined by several aromatic residues. To investigate further, we selected eight potentially critical residues of Asc-1 for functional studies: Asn52, Tyr131, Ile138, Phe243, Phe250, Tyr253, Glu257, Tyr333, and Arg339 (Fig. S5). Mutations of each residue to Ala in Asc-1 were generated by site-directed mutagenesis and assessed through the uptake of [14C]Glycine. Using LC-MS, we recently determined that Asc-1 predominantly works as an antiporter and that glycine is its preferred substrate[18]. To analyze wild-type and mutant activity, we expressed 4F2hc and Asc-1 in *Xenopus laevis* oocytes and determined the uptake of 100 μM [14C]Glycine (Fig. 3b), which occurs in exchange against endogenous amino acids.

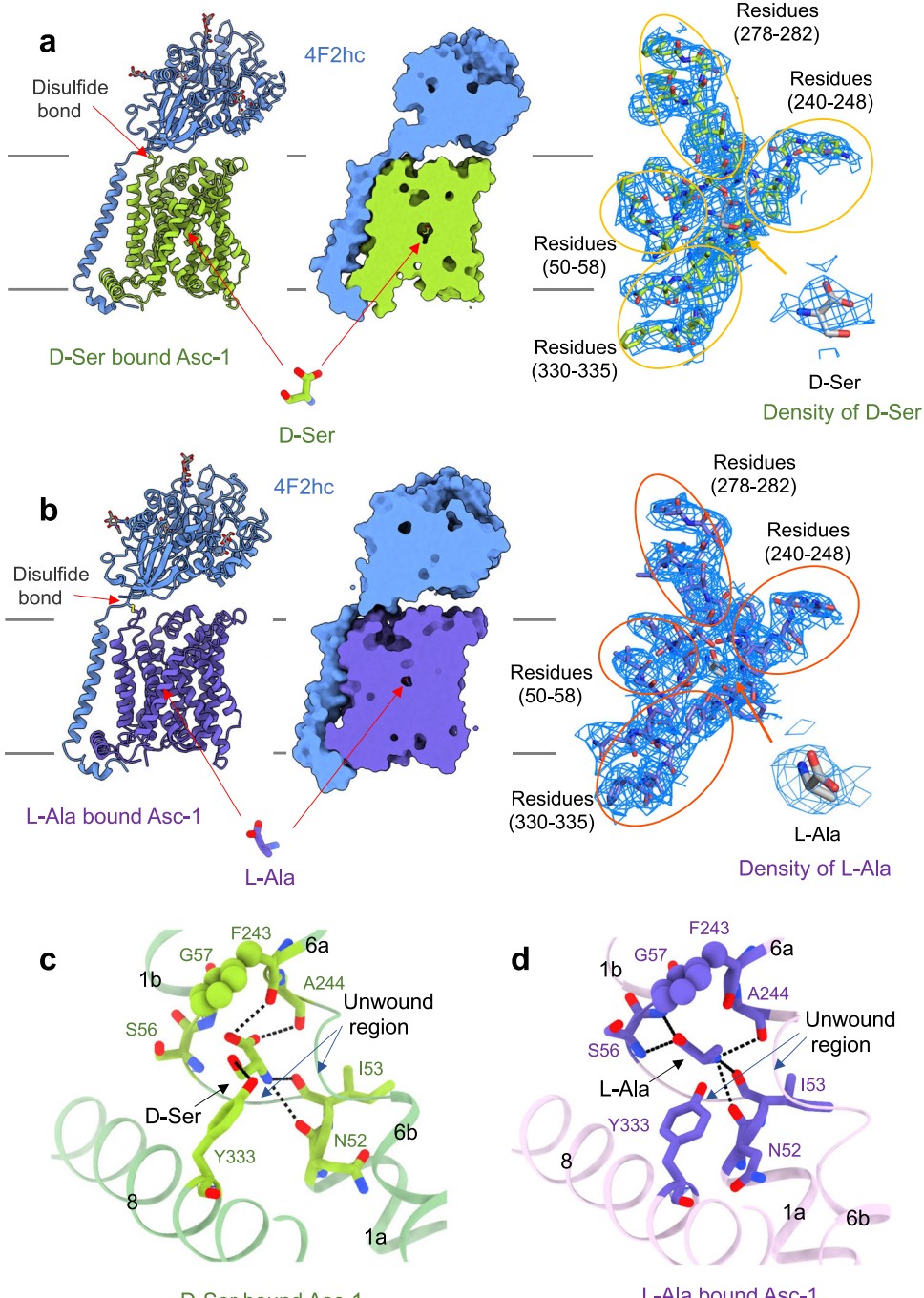

**Fig. 1 | Cryo-EM analysis of Asc-1-4F2hc complex. a** *Left:* Structure of the Asc-1-4F2hc in complex with D-Ser. The glycosylation moieties are shown as sticks. *Middle:* Asc-1 adopts an inward open conformation with D-Ser bound in the center of the substrate binding pocket. *Right:* Electron density maps of the substrate binding site and of D-Ser are shown at threshold of 7 σ. **b** *Left:* Structure of the Asc-1-4F2hc in complex with L-Ala. The glycosylation moieties are shown as sticks. *Middle:* Asc-1 adopts an occluded conformation with L-Ala bound in the center of the substrate binding pocket. *Right:* Electron density maps of the substrate binding site and of L-Ala are shown at threshold of 7 σ. The interaction network of D-Ser **c** and L-Ala **d** with Asc-1 are shown. 4F2hc is colored blue and Asc-1 bound D-Ser and Asc-1 bound L-Ala are colored green and purple. Dashed lines indicate plausible H-bond interactions.

Phe243 corresponds to residue Phe252 in LAT1, a critical gating residue in the transport cycle[7,35]. The F243A mutant exhibits nearly identical transport activity compared to the negative control, underscoring the pivotal role of Phe243. Notably, E257A and R339A mutants also display almost no transport activity, indicating that the interaction between Glu257 and Arg339 is crucial for the transport cycle. Consistently, we identify Glu257 and Arg339 as the intracellular gate of Asc-1. Additionally, N52A and Ile138A mutants exhibit a minor decrease in transport activity, suggesting a more subtle role in substrate

coordination or the transport cycle. Y131A, Phe250A, and Y333A mutants showed almost no decrease in transport activity. Given that we measured only the uptake of glycine, these three residues may play a role in preventing the entry of larger hydrophobic amino acids (Fig. 3b).

## Conformational changes induced by substrate binding

Substrate binding plays a pivotal role in triggering conformational changes in transporters. When aligning the D-Ser bound structure with

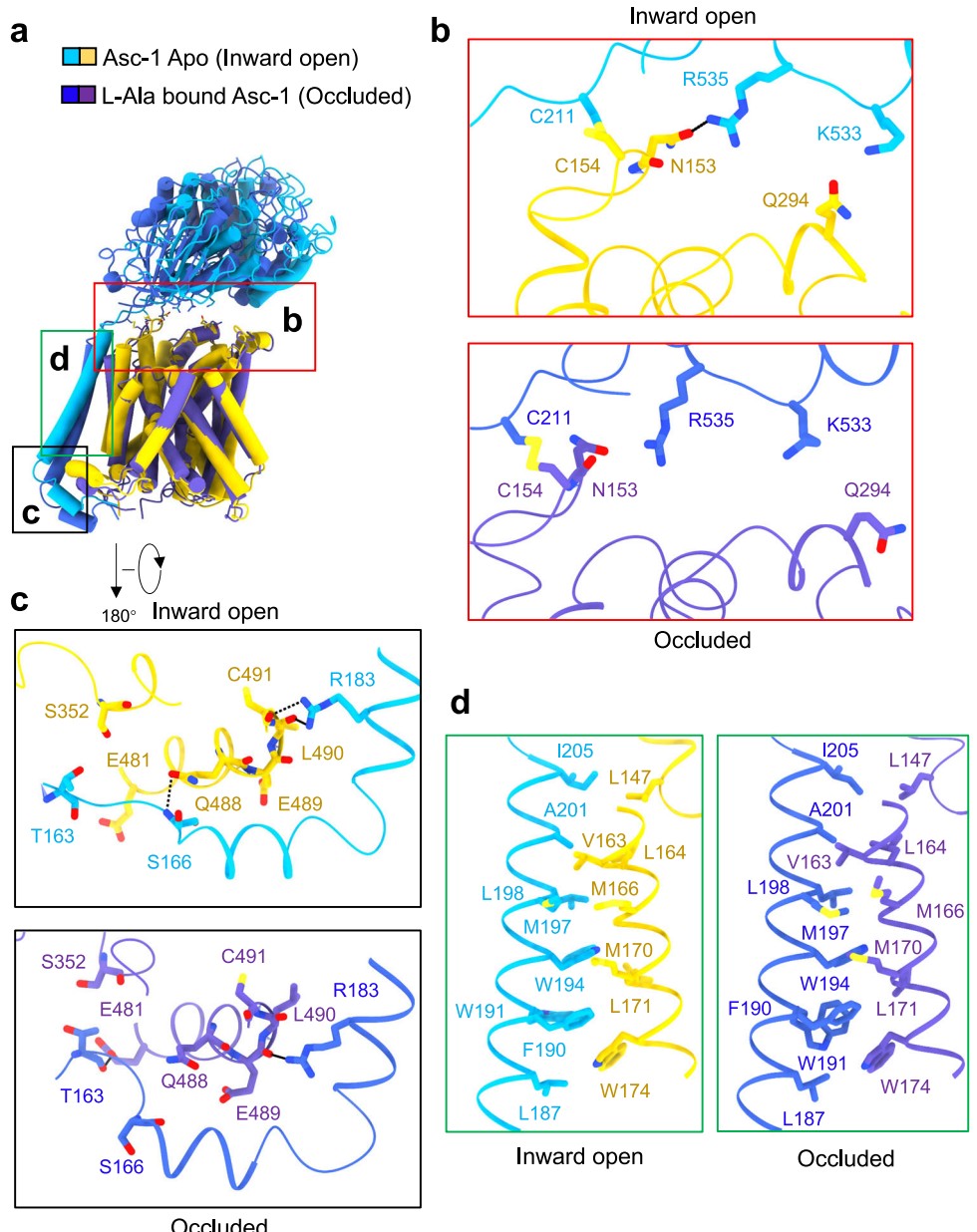

**Fig. 2 | Interactions between Asc-1 and 4F2hc during conformational change.**
**a** Structural comparison between Asc-1-4F2hc adopting different conformations.
When the substrate is bound to Asc-1, its state will change from inward open to
occluded state. The colored boxes correspond to detailed images in **b**, **c** and **d**.
**b** The extracellular interface between Asc-1 and 4F2hc. **c** The intracellular interface.
**d** The hydrophobic interactions between the transmembrane region of 4F2hc and
TM4 of Asc-1. Dashed lines indicate possible H-bond interactions.

the apo state in Asc-1, only a slight movement of TM1a towards the
interior is observed (Fig. 4a, b). However, upon further comparison
between the D-Ser and L-Ala bound structures, or apo and L-Ala bound
structures, significant movements are evident in TM1, TM6, and TM8
(Fig. 4c, d and Fig. S6a, b). The binding of L-Ala induces dramatic
conformational changes in the Asc-1-4F2hc + L-Ala complex when
compared to the D-Ser bound state. Specifically, Glu257 of the TM6b
tail rotates to form a salt bridge with Arg339 in TM8, further triggering
the movement of TM1a and resulting in the closure of the inward
transport vestibule (Fig. 4d).

To understand the difference between L-Ala and D-Ser, we
expressed Asc-1 and 4F2hc in *Xenopus laevis* oocytes and preloaded
oocytes with [14C]Glycine. Subsequently, exchange was induced by
addition of amino acids (1 mM) to the supernatant. D-Ser was found to
be a better trans-stimulating substrate than L-Ala (Fig. 4e). This

suggests that L-Ala binds more tightly to the transporter, slowing down
its turnover. Net efflux (facilitated diffusion) of [14C]Glycine in the
absence of extracellular substrate (NMDG-ND96 buffer) was low
compared to exchange. There was no difference between efflux
induced by L-Ser or D-Ser. Glycine was a relatively poor exchange
substrate and efflux induced by the non-substrate L-Leucine (L-Leu)
was close to that of ND96.

We also compared the occluded conformation of Asc-1 with that
of LAT1, including the JX-075 bound structure and Diiodo-Tyr bound
structure[35]. The position of the gating residue Phe243 in Asc-1 is similar
to Phe252 in the JX-075 bound structure of LAT1, indicating a con-
served transport mechanism among HAT members (Fig. S6c, d).
However, the Diiodo-Tyr bound structure of LAT1 exhibits a more
outward-open state, which might represent the next step in the Asc-1
transport cycle (Fig. S6e, f).

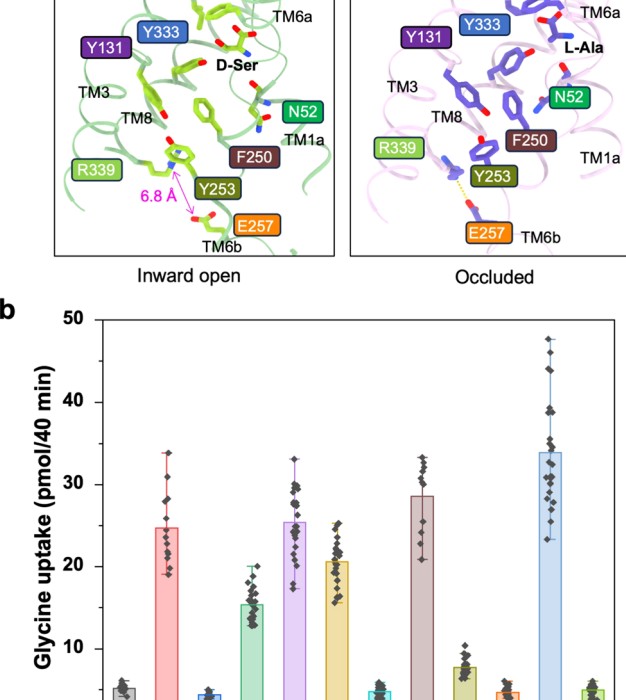

**Fig. 3 | The transport path of Asc-1. a** The substrate interface of Asc-1 in complex with D-Ser (Left panel) and L-Ala (Right panel). Glu257 forms a salt bridge with Arg339 in the occluded conformation. **b** Oocytes were injected with 10 ng cRNA of human 4F2hc plus 10 ng of human wildtype asc-1, *asc-1* mutants. As controls, oocytes were injected with h4F2hc alone (lane 1) or remained non-injected (ni). After incubation of 5 days uptake of 100 μM [$^{14}$C]glycine was determined over a time period of 40 min. Data are presented as mean ± SD, which was calculated from the uptake activity of independent oocytes (in the order as shown in Fig. 3b $n$ = 12, 13, 13, 24, 23, 26, 28, 11, 26, 28, 24, 26) derived from two independent batches.

## Discussion

In the working cycle of LeuT-fold transporters, the core domain (TM1, TM2, TM6, and TM7) typically undergoes a rocking motion between the outward-facing and inward-facing conformations[7,45–50]. In this study, the reported structures of different conformations of the Asc-1-4F2hc complex greatly expand the mechanistic understanding of human HAT transporters. The binding of L-Ala in the traditional substrate binding site of Asc-1 induces conformational changes in the core domain, resulting in the closure of the intracellular vestibule. This occluded conformation is very similar to the LAT1 conformation when bound to inhibitor JX-075. However, the question arises as to why D-Ser binding does not trigger a similar conformational change in Asc-1. The two ligands adopt slightly different poses, which may occur as a result of the stereochemical differences. L-Ala binding is positioned closer to the unwound region of TM1, potentially making it easier to trigger the conformational change compared to D-Ser. Consistently, the $K_M$-value of D-Ser for Asc-1 is higher than that of L-Ala[16]. Our efflux experiment suggest that this slows down the turnover of the transporter in the case of L-Ala. Thus, the conformation with the lowest energy adopted by the binding of the two amino acids could be slightly different. However, we currently lack a precise understanding by what mechanism these similar substrates lead to the stabilization of distinct conformations. Future Molecular dynamics simulations could be helpful to provide further insight.

Our structural studies identified two crucial intracellular gating residues, Glu257 and Arg339, within the Asc-1 transporter. Notably, these two residues are conserved across all HAT family members (Fig. S5). The corresponding residues, Glu266 and Arg348, in LAT1 can also form a stable salt bridge[35]. Notably, there is no salt bridge at the extracellular side of these HATs. Building upon these findings, we propose a unified working model for HAT transporters (Fig. 5).

In the absence of substrate binding, the transporter adopts an inward-facing conformation. A very similar conformation is adopted by the transporter when binding D-Ser. Upon binding of the amino acid substrate to the substrate binding pocket, it triggers the movement of TM1a, leading to the closure of the intracellular gate by forming the Glu-Arg salt bridge. Subsequently, the extracellular gating residue, usually a Phe residue, gradually shifts to an open position, resulting in the release of the amino acid substrate. Finally, another substrate can bind to the outward-facing state, allowing the transport cycle to restart (Fig. 5). Collectively, the structural determinations and in-depth analysis presented in this study mark a significant step towards achieving a comprehensive mechanistic understanding of human HAT family members and related diseases.

## Methods

### Protein expression and purification

The full-length human cDNA of *asc-1* (accession number: NM 019849.3) was subcloned into pCAG with N-terminal FLAG tag and *slc3a2* (isoform b, accession number: NM 001012662.2) into pCAG with N-terminal 10xHis tag. The mutations were generated by a standard two-step PCR. To ensure high tranfection rates, all plasmids were isolated using the GoldHi Endo Free Plasmid Maxi Kit (CWBIO).

The recombinant protein was produced through overexpression in HEK293F mammalian cells at 37 °C using a Multitron-Pro shaker (Infors), at a consistent speed of 130 rpm and an environment of 5% $CO_2$. To achieve co-expression of Asc-1 and 4F2hc, cells were transiently transfected with both plasmids using polyethyleneimines (PEIs) (YEASEN). Transfection was initiated when the cell density reached approximately $2.0 \times 10^6$/mL. For the transfection of one liter of cell culture, around 0.75 mg of Asc-1 plasmids and 0.75 mg of 4F2hc plasmids were pre-mixed with 3 mg of PEIs in 50 ml of fresh medium, allowing them to interact for 15 min before being introduced into the cell culture. After transfection for 60 h, cells were collected by centrifugation at 3500 × g for 15 min. Subsequently, cells were resuspended in a buffer comprising 25 mM Hepes (pH 7.5), 150 mM NaCl, supplemented with three protease inhibitors: aprotinin (1.3 μg/mL, AMRESCO), pepstatin (0.7 μg/mL, AMRESCO), and leupeptin (5 μg/mL, AMRESCO).

The extraction and purification of the Asc-1-4F2hc complex were carried out following established methods. First, the membrane fraction was solubilized at 4 °C for 2 h using 1% (w/v) Lauryl maltose neopentyl glycol (LMNG, Anatrace) supplemented with 0.1% (w/v) cholesteryl hemisuccinate Tris salt (Anatrace). Subsequently, cell debris was removed via centrifugation at 15,000 × g for 45 min. The resulting supernatant was then applied to anti-FLAG M2 affinity resin (Genscript). Following this step, the resin was thoroughly rinsed with a wash buffer that consisted of 25 mM Hepes (pH 7.5), 150 mM NaCl, and 0.02% (w/v) Glyco-diosgenin (GDN). The protein of interest was subsequently eluted using the wash buffer supplemented with 0.2 mg/mL FLAG peptide. The anti-FLAG M2 purified protein was further purified using a Ni-NTA affinity resin (Qiagen). Wash buffer and elution buffer of Ni-NTA resin was the wash buffer mentioned above plus 10 mM and 300 mM imidazole respectively. As a final step, the protein complex was subjected to size-exclusion chromatography (Superose 6 Increase 10/300 GL, GE Healthcare) in buffer containing 25 mM Hepes (pH 7.5), 150 mM NaCl and 0.02% GDN. The peak fractions were collected and concentrated for EM analysis.

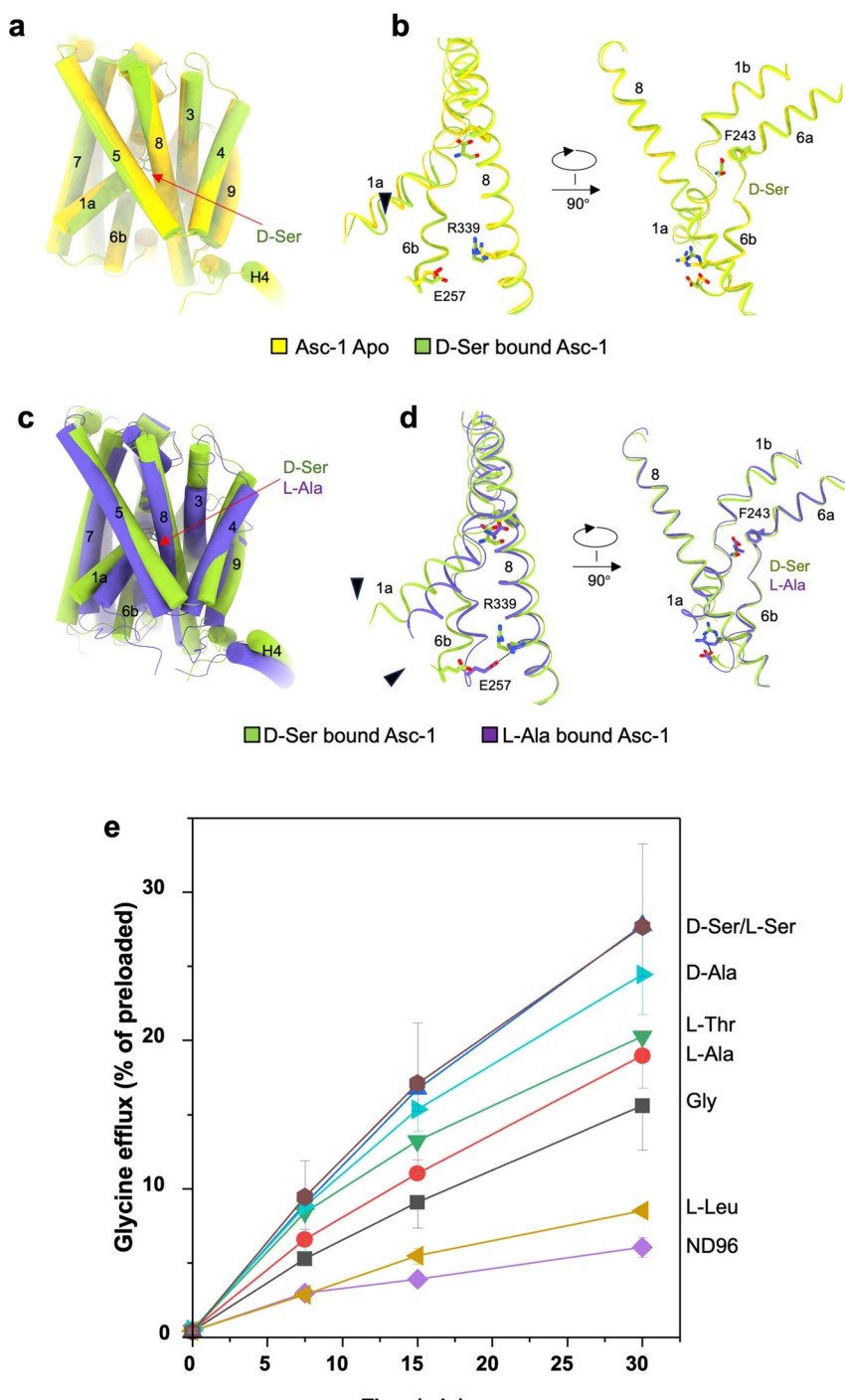

**Fig. 4 | The conformational change of Asc-1 during transport cycle. a, b** Structure comparison of Asc-1 apo and Asc-1 in complex with D-Ser shows only a slight movement of TM1a with D-Ser bound. **c, d** Structure comparison shows that Asc-1 in complex with D-Ser or L-Ala adopts different conformations. TM1a and TM6b are notably different. In the occluded conformation TM1a and TM6b are stabilized by a salt bridge between Glu257 and Arg339. Asc-1 apo and Asc-1 in complex with D-Ser adopt an inward open conformation while Asc-1 in complex with L-Ala adopts an occluded conformation. **e** The efflux transport assay of Asc-1-4F2hc complex. Oocytes were injected with 15 ng cRNA of human 4F2hc plus 15 ng human wild-type *asc-1*. After incubation of 5 days oocytes were preloaded with of 10 μM [$^{14}$C]glycine for one hour. Subsequently oocytes were washed and efflux induced by addition of 1 mM of the listed amino acids. Net efflux was determined in NMDG-ND96 buffer. Data are presented as mean values ± SD of three experiments each with $n = 8$ oocytes.

## Cryo-EM sample preparation and data acquisition

The Asc-1-4F2hc complex was incubated with 1 mM Ala or 1 mM D-Ser. Protein mixtures were concentrated to 10 mg/mL and aliquots (3.5 μl) of the mixture were placed on glow-discharged holey carbon grids (Quantifoil Au R1.2/1.3), which were blotted for 3.0 s or 3.5 s and flash-frozen in liquid ethane cooled by liquid nitrogen with the Vitrobot (Mark IV, Thermo Fisher Scientific). The cryo grids were transferred to a Titan Krios operating at 300 kV equipped with a Gatan K3 Summit detector and GIF Quantum energy filter. Movie stacks were automatically collected using AutoEMation[51], with a slit width of 20 eV on the energy filter and a defocus range from −1.4 μm to −1.8 μm in super-resolution mode at a nominal magnification of 81,000 ×. Each stack

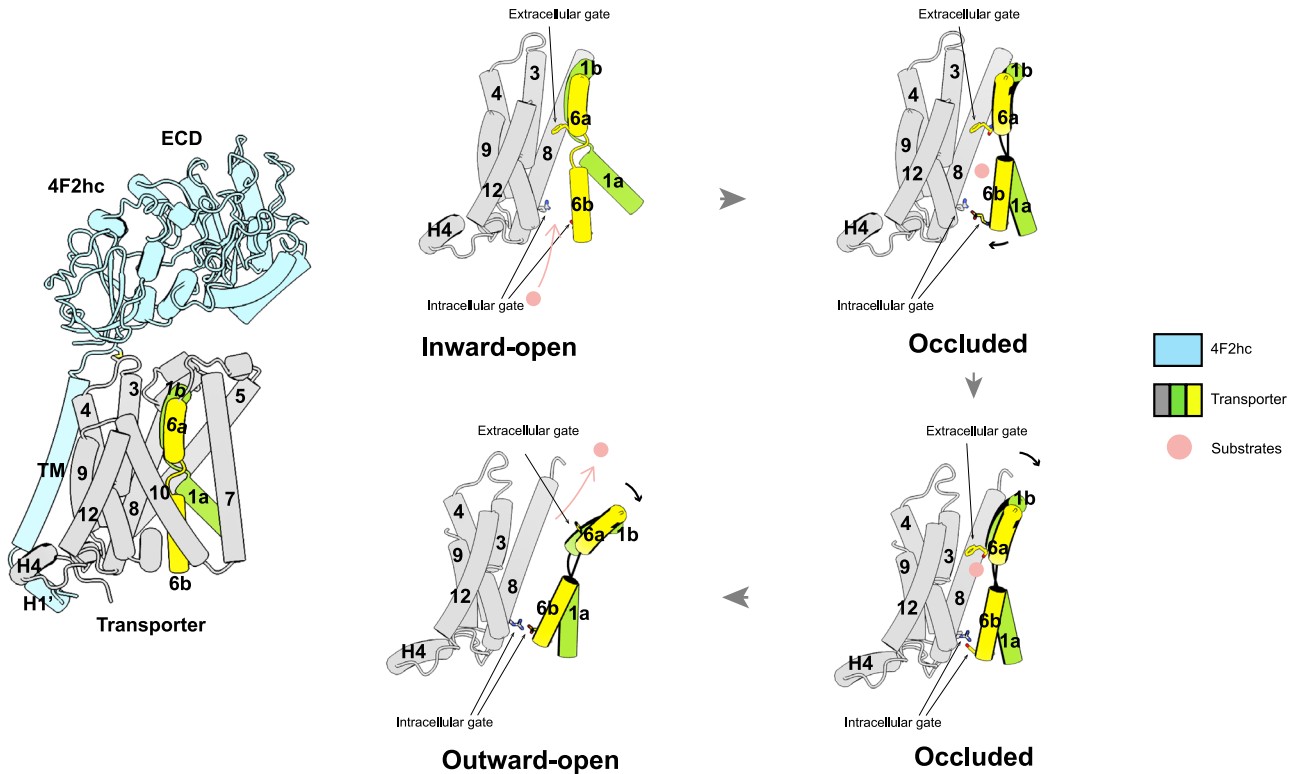

**Fig. 5 | Putative working model for the HAT complex.** The model shows a plausible transport cycle of heteromeric amino acid transporters (HAT). 4F2hc is colored blue and the transporter is colored grey, gold and green. The red spheres stand for substrates. Gate residues are shown by sticks.

was exposed for 2.56 s with an exposure time of 0.08 s per frame, resulting in a total of 32 frames per stack. The total dose rate was approximately 50 e⁻/Å² for each stack. The stacks were motion corrected with MotionCor2 [ref. 52] and binned twofold, resulting in a pixel size of 1.087 Å/pixel or 1.095 Å/pix. Meanwhile, dose weighting was performed[53]. The defocus values were estimated with Gctf[54].

### Data processing
The Cryo-EM structures were solved with cryoSPARC[55]. Patch-based CTF estimation was used to estimate the CTF correction parameters of micrographs in cryoSPARC. Particles were automatically picked using Templater picker. Several rounds of 2D classification were performed and the selected particles from 2D classification were subject to several cycles of heterogeneous refinement with C1 symmetry. Particles from the best class were subjected to Non-uniform Refinement, local CTF refinement and Non-uniform Refinement. The resolution was estimated with the gold-standard Fourier shell correlation 0.143 criterion[56] with high-resolution noise substitution[57]. Refer to Materials and Methods, Figs. S1–S3 and Table S1 for details of data collection and processing.

### Model building and structure refinement
The model building was accomplished with Phenix[58] and Coot[59]. The alphafold predict atomic model of the Asc1-4F2hc were used for the initial model of Asc-1-4F2hc bound with Ala and Asc-1-4F2hc bound with D-Ser and fitted into focused refined maps of substrate binding part using MDFF (molecular dynamics flexible fitting)[60]. Each residue was manually checked with Coot with the chemical properties taken into consideration during model building. Statistics associated with data collection, 3D reconstruction and model building is summarized in Table S1.

### Animals and ethical statement
Holding of *Xenopus laevis* frogs (purchased from Nasco, Fort Atkinson, WI) and the surgical procedure to remove parts of the ovary were approved by the Animal experimentation ethics committee of the Australian National University (Protocol A2020/28). All procedures were carried out in accordance with the recommendations of the Australian code for the care and use of animals for scientific purposes.

### Oocyte expression systems and flux experiments
*Xenopus laevis* oocytes were isolated and maintained as described previously[61]. Briefly, lumps of the ovary are removed under anesthesia (Approved protocol A2020/28) and digested using collagenase B (0.5 mg/mL) overnight in oocyte Ringer OR2- (82.5 mM NaCl), 2.5 mM KCl, 1 mM CaCl₂, 1 mM Na₂HPO₄, 5 mM HEPES; titrated with NaOH to pH 7.8. On the next day oocytes are washed extensively with OR2+ (OR2- plus 1 mM CaCl₂) and healthy stage 5/6 oocytes selected for cRNA injection. After addition of Gentamycin (10 mg/L) oocytes can be stored in OR2+ for up to seven days.

All constructs were transcribed in vitro *using* a T7 mMessage mMachine Kit for capped cRNA (Thermo Fisher Scientific). Selected oocytes were injected with 10 ng of human Asc-1 cRNA (or its mutants), 10 ng human 4F2hc cRNA and incubated for 5 days. Subsequently, uptake experiments were performed using NMDG-ND96 buffer (96 mM NMDG-Cl, 2 mM KCl, 1.8 mM CaCl₂, 1 mM MgCl₂, 5 mM HEPES; titrated with HCl to pH7.4). For uptake experiments NMDG-ND96 was supplemented with 100 µM [¹⁴C]glycine and incubated with the oocytes for 40 min. To terminate uptake, oocytes were washed three times with 4 ml of ice cold NMDG-ND96, transferred to scintillation vials and counted for determination of accumulated radioactivity. For efflux, oocytes were injected with 15 ng cRNA of human 4F2hc plus 15 ng human wild-type *asc-1*. After incubation for 5 days, groups of 8 oocytes were preloaded with of 10 µM [¹⁴C]glycine for one hour. Subsequently oocytes were washed and efflux induced by addition of 1 mL NMDG-ND96 buffer containing 1 mM extracellular amino acids. Aliquots of 0.2 mL were taken at different time points for the determination of radioactivity.

## Site-directed mutagenesis

Human Asc-1 cloned into oocyte expression vector pGEM-He-Juel[18] was used as a template for mutagenesis. Mutations were introduced using the QuikChange XL Site-Directed Mutagenesis kit (Agilent #200517). All mutations were confirmed by Sanger sequencing. The following primers were used (only forward primers shown, inverted sequence for reverse primer):

N52A: ACCATCATCATCGGGGCCATCATCGGCTCGGGC
Y131A: GCCGTCCTCATCATGGCCCCCACCAGCCTTGCT
I138A: ACCAGCCTTGCTGTCGCCTCCATGACCTTCTCC
F243A: TTCCTCCAGGGCTCCGCCGCCTTCAGTGGCTGG
F250A: TTCAGTGGCTGGAACGGCCTCAACTATGTCACC
Y253A: TGGAACTTCCTCAACGCTGTCACCGAGGAGATG
E257A: AACTATGTCACCGAGGCGATGGTTGACGCCCGA
Y333A: GGAGGGATCAATGGTGCCCTGTTCACCTACTCC
R339A: CTGTTCACCTACTCCGCGCTGTGCTTCTCTGGA

## Reporting summary

Further information on research design is available in the Nature Portfolio Reporting Summary linked to this article.

## Data availability

Cryo-EM maps and molecular models have been deposited in the Electron Microscopy Data Bank (EMDB) and Protein Data Bank (PDB), respectively. Accession codes are listed here and in Supplementary Table 1. The cryo-EM maps have been deposited in the Electron Microscopy Data Bank (EMDB) under accession codes EMD-37671 (Asc-1-4F2hc complex in apo state); EMD-37672 (Asc-1-4F2hc complex in L-Ala bound state); and EMD-37675 (Asc-1-4F2hc complex in D-Ser bound state). The atomic coordinates have been deposited in the Protein Data Bank (PDB) under accession codes PDB: 8WNS (Asc-1-4F2hc complex in apo state); PDB: 8WNT (Asc-1-4F2hc complex in L-Ala bound state); and PDB: 8WNY (Asc-1-4F2hc complex in D-Ser bound state). All other data will be made available upon request. The source data underlying Figs. 3b and 4e are provided as a Source Data file. Source data are provided with this paper.

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

## Acknowledgements

We thank the Cryo-EM Facility of Southern University of Science and Technology (SUSTech) and Westlake University for providing the facility support. We thank Shuman Xu, Lei Zhang, and Peiyao Li at the Cryo-EM Center of SUSTech for technical support in electron microscopy data acquisition. Renhong Yan is an investigator of SUSTech Institute for Biological Electron Microscopy. We thank Zhenyuan Liu for technical support on computing environment. We thank Aditya Yadav for her help with the site-directed mutagenesis. This work was funded by the National Natural Science Foundation of China (82202517 to R.Y.), the Major Talent Recruitment Program of Guangdong Province (2021QNO2Y167 to R.Y.), and the Natural Science Foundation of Guangdong Province (The transport mechanism of human amino acid transporter complex ASC-1-4F2hc to Y.G.).

## Author contributions

R.Y. conceived the study. R.Y., S.B., Y.L. and Y.G. designed the experiments. Y.L., R.Y. and L.D. did the molecular cloning and protein purification. Y.L. and R.Y. did the Cryo-EM data collection and processing. Y.L. and Y.G. built and refined structural models. A.B. did site directed mutagenesis experiments, sequencing and *Xenopus laevis* oocytes uptake and efflux assays. All authors analyzed the results. R.Y. and S.B. wrote the manuscript.

## Competing interests

The authors declare no competing interests.
