## [Peer Review File · Nature Communications]

Cryo-EM structure of the human Asc-1 transporter complexReviewer #1 (Remarks to the Author):

Yaning Li et al., Stefan Bröer & Renhong Yan present cryo-EM structures and functional studies of the Asc1 amino acid transporter (also known as SLC7A10), which forms a CD98 complex with the "heavy subunit" 4F2hc, also known as CD98hc (or SLC3A2), stabilized by an intermolecular disulphide bridge. This transporter complex can operate both through exchange (which is typical of the heteromeric amino acid transporters HAT) and outward-oriented facilitated transport (which is unique to Asc1). They report cryo-EM structures at 3.4-3.6 Å overall resolution for the apo-form and complexes with D-Ser and L-Ala. Through structural analysis of binding sites and conformational changes and mutational studies they reach a mechanistic model of transport. They find a particular role of the Tyr333 residue and adaptation to D-Ser vs. L-Ala substrates (inward-open vs. occluded, respectively). This is an exciting study of an important transporter system. The manuscript is written in a concise style, and is easy to read. Below are some points that may improve the manuscript

1. First of all, there is no such thing as L-Gly – it is simply glycine/Gly, with no chiral centers
2. " In the nervous system, L-Gly and D-Ser serve as two endogenous glutamate co-agonists that activate N-methyl-d-aspartate (NMDA) receptors by binding to the strychnine insensitive binding site, making Asc-1 a promising druggable target for treating cognitive affections impairment and schizophrenia diseases¹⁹⁻²⁴ " – please be a bit more specific on how Asc1 might be drug target – activating or inhibiting transport?
3. For the "Structural determination of Asc-1-4F2hc complex" section – please include a brief note on how the membrane protein complex was solubilized and purified regarding detergent, lipids, cholesterol.
4. D-Ser and L-Ala seemingly stabilize the transporter in two different states, inward-open and inward-occluded. Are they also associated with two different transport modes, exchange and facilitated transport?
5. For consistency with the limited resolution of the study, please refrain from using specific references of hydrogens bonds
6. The balance between the main article and supplementary material is not ideal – please consider moving parts of the supplementary material to the main article

Minor points

7. " Substrate binding plays a pivotal role in triggering conformational changes in the Asc- 1 transporter. " – this is generally the case for transporters
8. Fig. 4 labels for TM1 and 6 helices are interchanged
9. The density for D-Ser is less convincing qlthog substrate is certainly there. Is the D-Ser sample pure or will it contain a significant trace of L-Ser or other amino acids binding at higher apparent affinity?

Reviewer #2 (Remarks to the Author):

The manuscript by Li et al describes the cryo-EM structure of the human Asc-1 transporter complex. This transporter is similar to another amino acid transporter, LAT1. The Asc-1 transporter complex transports amino acids, including L-Ala, L-Ser, Gly, and D-Ser, as potential regulators of NMDA receptors. The involvement of NMDA receptors in various cognitive functions and neuropsychiatric conditions has been well established, boosting the current study's significance. In this study, the cryo-EM structures of the Asc-1-4F2hc complex were determined in three states: apo, D-Ser bound, and L-Ala bound, with resolutions of 3.6 Å, 3.5 Å, and 3.4 Å respectively. The apo and D-Ser-bound states display an inward open conformation, while the L-Ala-bound state adopts an occluded conformation. Extensive interactions between 4F2hc and Asc-1 were identified across multiple regions, similar to the LAT1-4F2hc complex, with a notable conserved disulfide bond and specific hydrogen bonds. Differences in interaction mechanisms compared to LAT1-4F2hc were observed, especially in the apo

state where Asc-1 forms hydrogen bonds with 4F2hc, which are altered upon L-Ala binding, leading to a remodeled intracellular interface. The binding patterns of D-Ser and L-Ala within the Asc-1-4F2hc complex differ slightly, with each amino acid forming unique hydrogen bonds with specific residues. The transport path of Asc-1, identified by comparing the D-Ser and L-Ala bound conformations, is lined with several aromatic residues and involves critical interactions between Glu257 and Arg339, which are disrupted in the inward-facing state. The role of specific residues in transport activity was highlighted, with Phe243 being a critical gating residue and Glu257 and Arg339 identified as the intracellular gate of Asc-1. Mutations in critical residues revealed varied impacts on transport activity. Substrate binding was found to be crucial for triggering conformational changes in Asc-1, with comparisons between different bound structures revealing significant movements in TM1, TM6, and TM8. Particularly, L-Ala binding induces substantial changes, leading to the closure of the inward transport vestibule. Finally, a comparison with the LAT1 transporter suggests a conserved transport mechanism among HAT members despite differences in their conformational states. As summarized above, the authors made a significant stride toward understanding this important amino acid transporter. These structures demonstrate substrate-specific conformational alterations essential for functions. I recommend this paper to be published upon addressing the points below.

Major points

-Substrate bindings (Fig. 1 and Fig. S4). While the local protein conformational change specific to the L-Ala and D-Ser appears convincing, the binding poses are not. The cryo-EM density in Fig. S4 should not guide you to claim differences in the binding modes between D-ser and L-Ala. It should be noted in the manuscript that the cryo-EM density did not resolve sufficiently to determine the binding modes precisely and that the binding pose is what the authors tentatively suggest based on the not well-resolved density.

-In Fig. 1c and d, are the yellow dots from the carboxylate group of Ala244 connected to the methyl group of L-Ala? If so, the dots should be removed. Furthermore, the authors should only show dots within 3.2 angstroms for polar interactions.

-The L-Gly uptake assay. Is it possible to also assess efflux by this assay by injecting [3H]Gly into oocytes and analyzing the ND96 media?

-The authors should explain what the oocyte uptake experiments are briefly in the main text (e.g., uptake of [H3]Gly. Explain why the authors chose uptake, not efflux).

-What is 'ni' in Fig. 2. The authors could put more information into the figure legends.

-What could be the functional role of 4F2hc? The functional-state-dependent interactions with the Asc-1 indicate some involvement in regulating functions. The authors could conduct site-directed mutagenesis on the Asc-1-4F2hc interface. They could do this for this manuscript or in their future work.

-“Discussion” does not have much detail and is a bit confusing. For example, “Consistently, the affinity of D-Ser for Asc-1 is lower than that of L-Ala.” Did the authors measure the affinity of D-Ser and L-Ala? If V_{max} does not change, then does the kinetics of transport change? In other words, what is the slope of the plot if you’ve recorded transporter activities at multiple time points? Something must change if there is such a dramatic preference for L-Ala to induce the occluded state. The entire argument in this argument is obscure.

-Map vs. Figure/interpretation

Unless there are prior knowledge, assessing the binding poses of Ala and D-serine, as presented in this study, would be challenging.

Several concerns arise regarding the accuracy of substrate placement. Claiming that they bind differently based solely on the current map quality may be an exaggeration. Furthermore, there are concerns about the authors' interpretation of their models. In the case of Ala, the proximity of the carboxylate oxygens (OXT) to the Phe243 ring (3.18 Å) is noteworthy. Additionally, these carboxylate oxygens do not appear to engage in polar interactions with the hydroxyl group of Tyr333, given the distance of 4.05 Å. For D-Ser, similar observations can be made. The carboxylate oxygen (O) seems not to form polar interactions with the mainchain nitrogen of Ser56 and Gly57, with distances of 4.16 Å and 3.63 Å, respectively.

Minor pts

-Three-letter or one-letter amino acid display. The authors need to be consistent.

-Where is "the unwound region of TM1 and TM6 (Fig. 1c)." The figure does not guide a reader to see unwinding.

-" Notably, the side chain of substrate D-Serin can..." D-Serine.

-" L-Gly and D-Ser serve as two endogenous glutamate co-agonists that activate N-methyl-d-aspartate (NMDA) receptors by binding to the strychnine insensitive binding site, making Asc-1 a promising druggable target for treating cognitive affections impairment and schizophrenia diseases"

Hansen et al (Pharmacol Rev. 2021 Oct;73(4):298-487. doi: 10.1124/pharmrev.120.000131.)

Furukawa and Gouaux (EMBO J . 2003 Jun 16;22(12):2873-85. doi: 10.1093/emboj/cdg303.)

These two papers are appropriate to be cited here.

Response to Reviewers

We appreciate the insightful and constructive comments from both reviewers. We have performed additional experiments to address all comments. Below please find our point-to-point responses.

Reviewer #1:

We greatly appreciate the reviewer's recognition of the significance of our study. He or she made a number of very constructive comments. We appreciate this reviewer's input and have revised the manuscript accordingly. Below please find our point-to-point response to his/her concerns:

1. First of all, there is no such thing as L-Gly – it is simply glycine/Gly, with no chiral centers

We apologize for this mistake. It has been corrected in the revision.

2. " In the nervous system, L-Gly and D-Ser serve as two endogenous glutamate co-agonists that activate N-methyl-d-aspartate (NMDA) receptors by binding to the strychnine insensitive binding site, making Asc-1 a promising druggable target for treating cognitive affections impairment and schizophrenia diseases¹⁹⁻²⁴ " – please be a bit more specific on how Asc1 might be drug target – activating or inhibiting transport?

We appreciate this insightful comment. Unfortunately, the literature is ambiguous, because Asc-1 has different roles in glycinergic neurons versus glutamatergic neurons. Inhibiting transport (i.e. the release of D-Ser) has been proposed for the treatment of schizophrenia but inhibition may also cause hyperexcitability due to reduced glycine recycling. We have clarified and extended this section in the revised introduction.

3. For the "Structural determination of Asc-1-4F2hc complex" section – please include a brief note on how the membrane protein complex was solubilized and purified regarding detergent, lipids, cholesterol.

We thank this reviewer for this suggestion. This part was modified as: *"The membrane fraction was solubilized using 25 mM Hepes (pH 7.5), 150 mM NaCl, 1% Lauryl maltose neopentyl glycol (LMNG) supplemented with 0.1% cholesteryl hemisuccinate Tris salt (CHS). After tandem affinity purification in modified solubilization buffer containing 0.01% LMNG and 0.001% CHS. During subsequent size exclusion chromatography (SEC) in 25 mM Hepes (pH 7.5), 150 mM NaCl, and 0.02% Glyco-diosgenin (GDN), the complex exhibited a single, well-defined peak and its quality was confirmed through Coomassie blue staining on sodium dodecyl sulfate–polyacrylamide gel electrophoresis (SDS–PAGE) gels, indicating a high level of homogeneity (Fig. S1a)."*

4. D-Ser and L-Ala seemingly stabilize the transporter in two different states,

inward-open and inward-occluded. Are they also associated with two different transport modes, exchange and facilitated transport?

We appreciate this insightful comment. D-Ser and L-Ala are both uptake and efflux substrates of Asc-1 and could stabilize the transporter in the inward-open and inward-occluded conformation, respectively. We propose that the different binding poses of L-Ala and D-Ser may account for this variation. L-Ala binding is positioned closer to the unwound region of TM1, potentially making it easier to trigger the conformational change compared to D-Ser. Consistently, the affinity of D-Ser for Asc-1 is lower than that of L-Ala. Thus, the conformation with the lowest energy adopted by the binding of the two amino acids is slightly different. As for the transport modes, we have added an efflux experiment to the manuscript and it supports the notion that L-Ala binds more tightly to the transporter. To study facilitated diffusion, we would need radiolabeled D-Ser, which is not readily available in Australia. A more detailed characterization of the transport mode of all substrates would go beyond the scope of this study.

5. For consistency with the limited resolution of the study, please refrain from using specific references of hydrogen bonds

Point taken. We have removed the specific references to hydrogen bonds in the revised Fig.1 and Fig. S5 (Fig.2 in the revision).

Fig.1 (revised Fig.1) The substrate binding mode in Asc-1

Fig.2 (originally Fig.S5) Interactions between Asc-1 and 4F2hc

6. *The balance between the main article and supplementary material is not ideal – please consider moving parts of the supplementary material to the main article*

We appreciate this comment. We moved Fig. S5 from the supplement to Fig. 2 and added the efflux assay result as Fig.4e, now the number of Supplementary figures is 6 and the number of main figures is 5.

Minor points

7. *” Substrate binding plays a pivotal role in triggering conformational changes in the Asc- 1 transporter. ” – this is generally the case for transporters*

Point taken. We revised this sentence as “*Substrate binding plays a pivotal role in triggering conformational changes in transporters*”.

8. Fig. 4 labels for TM1 and 6 helices are interchanged

.Thanks for spotting this mistake. We have corrected the labels (Revised Fig.5).

Fig.3 (revised Fig.5) The proposed working model of HAT family members

9. The density for D-Ser is less convincing qthog substrate is certainly there. Is the D-Ser sample pure or will it contain a significant trace of L-Ser or other amino acids binding at higher apparent affinity?

We thank this reviewer for this critical comment. We incubated our protein with 1 mM D-ser before preparing cryo-EM samples and the D-Ser substrate was purchased through Sigma-Aldrich company and is of analytical reagent quality (sigma, $\geq 98\%$ (TLC)). The density of D-Ser in the structure compared with the apo structure is obvious. Furthermore, the conformational change induced by D-Ser compared with apo structure and the L-Ala bound structure further supports specific binding of D-Ser. In conclusion, we are confident that it is D-Ser in complex with Asc-1-4F2hc.

Reviewer#2

We thank the reviewer for recognizing the significance of our study. We appreciate this reviewer for his/her input.

Major points

1.Substrate bindings (Fig. 1 and Fig. S4). While the local protein conformational change specific to the L-Ala and D-Ser appears convincing, the binding poses are not. The cryo-EM density in Fig. S4 should not guide you to claim differences in the binding modes between D-ser and L-Ala. It should be noted in the manuscript that

the cryo-EM density did not resolve sufficiently to determine the binding modes precisely and that the binding pose is what the authors tentatively suggest based on the not well-resolved density.

We thank this reviewer for this critical comment. We acknowledge that the current resolution of Asc-1 did not allow a precise fit of D-Ser and L-Ala. We analysed the interaction network of Asc-1 and its ligand and aligned our structure with homologous proteins from the HAT family and built similar pose of ligands. These information supports the substrate binding pose in our structures. We have modified the section to acknowledge the limitations of the modelling.

Fig.4 The aligned substrate binding modes in different HAT members

2. In Fig. 1c and d, are the yellow dots from the carboxylate group of Ala244 connected to the methyl group of L-Ala? If so, the dots should be removed. Furthermore, the authors should only show dots within 3.2 angstroms for polar interactions.

We thank this reviewer for this critical comment, which was also made by reviewer 1. As suggested, we removed references to hydrogens bonds in the figures (Please refer to the figure below).

Fig.5 (revised Fig.1) The substrate binding modes in Asc-1

3. The L-Gly uptake assay. Is it possible to also assess efflux by this assay by injecting [^3H]Gly into oocytes and analyzing the ND96 media?

We thank this reviewer for this comment. Yes, this experiment is possible. We prefer to preload with oocytes because not all injected radiolabel is accessible for efflux due to the dense egg yolk inside oocyte. The experiment is now shown as Fig. 4e.

4. The authors should explain what the oocyte uptake experiments are briefly in the main text (e.g., uptake of [^3H]Gly. Explain why the authors chose uptake, not efflux).

As suggested, we have added a brief description of the uptake assay in the revision. Uptake, as well as efflux can be measured. For convenience we used uptake experiments, which measure exchange against endogenous amino acids. We have added an efflux experiment using wild-type Asc-1 to demonstrate the exchange and net efflux mode (Please refer to the figure below).

Fig.6 (revised Fig.4e) The efflux transport assay for Asc-1-4F2hc complex

5. What is 'ni' in Fig. 2. The authors could put more information into the figure legends.

Apologies for the omission. A more detailed figure legend has been added in the revision, including the explanation of n.i. as non-injected oocytes

6. What could be the functional role of 4F2hc? The functional-state-dependent interactions with the Asc-1 indicate some involvement in regulating functions. The authors could conduct site-directed mutagenesis on the Asc-1-4F2hc interface. They could do this for this manuscript or in their future work.

Many thanks for raising this issue. The role of 4F2hc affecting the transport activity of the associated light subunits such as LAT1, Asc-1 and others has been investigated in our

previous study (Yan et al, *Nature*, 2019) and confirmed the critical functional role of 4F2hc. We appreciate this suggestion to investigate the influence of 4F2hc on Asc-1 but a thorough study of the functional role of 4F2hc would be an independent study in the future.

7. “Discussion” does not have much detail and is a bit confusing. For example, “Consistently, the affinity of D-Ser for Asc-1 is lower than that of L-Ala.” Did the authors measure the affinity of D-Ser and L-Ala? If V_{max} does not change, then does the kinetics of transport change? In other words, what is the slope of the plot if you’ve recorded transporter activities at multiple time points? Something must change if there is such a dramatic preference for L-Ala to induce the occluded state. The entire argument in this argument is obscure.

We thank this reviewer for this critical comment. To understand the differences between L-Ala and D-Ser, we have added an additional experiment using D-Ser and L-Ala as exchange substrates. D-Ser is a better efflux substrate than L-Ala, which is consistent with a stronger binding of L-Ala to the transporter. We have also edited the corresponding section in the discussion to improve clarity.

8. Map vs. Figure/interpretation.

Unless there are prior knowledge, assessing the binding poses of Ala and D-serine, as presented in this study, would be challenging. Several concerns arise regarding the accuracy of substrate placement. Claiming that they bind differently based solely on the current map quality may be an exaggeration. Furthermore, there are concerns about the authors' interpretation of their models. In the case of Ala, the proximity of the carboxylate oxygens (OXT) to the Phe243 ring (3.18 Å) is noteworthy. Additionally, these carboxylate oxygens do not appear to engage in polar interactions with the hydroxyl group of Tyr333, given the distance of 4.05 Å. For D-Ser, similar observations can be made. The carboxylate oxygen (O) seems not to form polar interactions with the mainchain nitrogen of Ser56 and Gly57, with distances of 4.16 Å and 3.63 Å, respectively.

We thank this reviewer for this critical comment. We have modified the structures now and aligned our structure with the homologous proteins from the HAT family and build similar poses of the ligands (Please refer to the figure below).

Fig.7 The aligned substrate binding modes in different HAT members

Minor points:

9. *Three-letter or one-letter amino acid display. The authors need to be consistent.*

We thank this reviewer for this critical comment. We used Three-letter amino acid display in the result section and one-letter amino acid display in the figures, which is consistent with previous publication style.

10. *Where is “the unwound region of TM1 and TM6 (Fig. 1c).” The figure does not guide a reader to see unwinding.*

We thank this reviewer for this critical comment. We have revised the Fig. 1.

Fig.7 (revised Fig.1) The highlighted unwound region of TM1 and TM6

11. *“Notably, the side chain of substrate D-Serin can...” D-Serine.*

Corrected.

12. *“L-Gly and D-Ser serve as two endogenous glutamate co-agonists that activate N-methyl-d-aspartate (NMDA) receptors by binding to the strychnine insensitive binding site, making Asc-1 a promising druggable target for treating cognitive affections impairment and schizophrenia diseases” Hansen et al (Pharmacol Rev. 2021 Oct;73(4):298-487. doi: 10.1124/pharmrev.120.000131.) Furukawa and Gouaux (EMBO J . 2003 Jun 16;22(12):2873-85. doi: 10.1093/emboj/cdg303.) These two papers are appropriate to be cited here.*

We thank this reviewer for this critical comment. We have added these references to the revision.

Reviewer #1 (Remarks to the Author):

The authors have in most cases responded well to comments and suggestions from the reviewers, and I have only few comments to the revised version.

1) I asked for less specific statements of hydrogen bonds, and they have done so carefully in the text. I think it is a pity not to indicate the interactions that are likely/potential/possible, so in my opinion it will be fine to include dashed lines for possible H-bond interactions if only distances are not considered accurate enough to discern that and figure legends are clear with the disclaimers

2) The substrate binding pockets of the L-Ala, D-Ser complexes do show density for binding of something, and most likely those added substrates - however, looking at Fig. S4 the density in particular for D-Ser is probably not very explicit regarding possible interactions and mechanisms, and it seems that the D-Ser side chain sticks out of density. L-Ala and D-Ser are also shown from complexes in different orientations it seems (a no-go) . Perhaps the substrate is bound in a flexible form under the given experimental conditions or perhaps structures represent mixed binding modes. Difficult to say at the given resolution and quality of the map, so specific interpretations on mechanisms must be cautious. Density for the substrate pockets should be shown in close-up and included in the main text and discussed - the indications of density for the bound substrates in Figure 1a+b are not useful and should be replaced with density mesh for closeup all-atom model representations. Show how things are and make the best possible interpretation of it. Instead, I'm left with the impression that the density is perhaps not as unambiguous as the model representation and analysis might indicate, and that would be a mistake.

Reviewer #2 (Remarks to the Author):

1. In discussing the D-Ser and L-Ala specific stabilization of the Asc-1 conformation, it is important for the authors to explicitly acknowledge their lack of understanding regarding the mechanism by which these similar substrates lead to the stabilization of distinct conformations.

2. Regarding substrate binding poses and orientations, the authors should clearly indicate that they are modeled based on the similarity of the binding pocket to that found in HAT. Perhaps address that MD simulations may be necessary to speculate deeper in the favorable poses.

3. In the comparison of uptake/intake assays, it is important for the authors to distinguish between potency and binding affinity explicitly. They should clarify that they are comparing differences in potency, not merely binding affinity.

Reviewer #1

1. I asked for less specific statements of hydrogen bonds, and they have done so carefully in the text. I think it is a pity not to indicate the interactions that are likely/potential/possible, so in my opinion it will be fine to include dashed lines for possible H-bond interactions if only distances are not considered accurate enough to discern that and figure legends are clear with the disclaimers.

Response: We thank this reviewer for this clarification. We recognize that at this resolution the substrate binding poses are tentative, so we initially removed all the hydrogen bonds. Following the suggestion we have added plausible hydrogen bonds within a distance of 3.2 angstrom and added disclaimers to the figure legends.

2. The substrate binding pockets of the L-Ala, D-Ser complexes do show density for binding of something, and most likely those added substrates - however, looking at Fig. S4 the density in particular for D-Ser is probably not very explicit regarding possible interactions and mechanisms, and it seems that the D-Ser side chain sticks out of density. L-Ala and D-Ser are also shown from complexes in different orientations it seems (a no-go). Perhaps the substrate is bound in a flexible form under the given experimental conditions or perhaps structures represent mixed binding modes. Difficult to say at the given resolution and quality of the map, so specific interpretations on mechanisms must be cautious. Density for the substrate pockets should be shown in close-up and included in the main text and discussed - the indications of density for the bound substrates in Figure 1a+b are not useful and should be replaced with density mesh for closeup all-atom model representations. Show how things are and make the best possible interpretation of it. Instead, I'm left with the impression that the density is perhaps not as unambiguous as the model representation and analysis might indicate, and that would be a mistake.

Response: We sincerely thank this reviewer for the insightful comments. We agree with that at this resolution, substrate poses are clearly tentative. The current poses are supported by homologous substrate orientations in other structures of the SLC7 family and are supported by docking experiments. It appears unlikely that the poses of L-Ala and D-Ser are exactly the same as they equilibrate into different conformations of the Asc-1 transporter. As suggested we have added the electron density maps to the main manuscript and rearranged Figure 1 accordingly.

Reviewer #2

1. In discussing the D-Ser and L-Ala specific stabilization of the Asc-1 conformation, it is important for the authors to explicitly acknowledge their lack of understanding regarding the mechanism by which these similar substrates lead to the stabilization of distinct conformations.

Response: We appreciate this suggestion and have added the following statement to the discussion: "*However, we currently lack a precise understanding by what mechanism these similar substrates lead to the stabilization of distinct conformations.*"

2. Regarding substrate binding poses and orientations, the authors should clearly indicate that they are modeled based on the similarity of the binding pocket to that found in HAT. Perhaps address that MD simulations may be necessary to speculate deeper in the favorable poses.

Response: Point taken. We have added this description in the result as “*Based on the analysis of the interaction between substrate and Asc-1, as well as the similarity of substrate-binding pockets among the HAT family homologous proteins^{41,45}, we built L-Ala and D-Ser in the density of substrates.*” We further added to the discussion: “*Future Molecular dynamics simulations could be helpful to provide further insight*”.

3. In the comparison of uptake/intake assays, it is important for the authors to distinguish between potency and binding affinity explicitly. They should clarify that they are comparing differences in potency, not merely binding affinity.

Response: We thank this reviewer for the suggestion. We have specified in the discussion: “*Consistently, the K_M -value of D-Ser for Asc-1 is higher than that of L-Ala.*” There is no other place in the manuscript where we used the term affinity.